# EASYSPIKING: SPIKE-FRIENDLY FUNCTION APPROXIMATIONS FOR SPIKING LLMS WITHOUT FINE-TUNING

## ABSTRACT

Transformer-scale large language models (LLMs) deliver state-of-the-art accuracy but demand heavy floating-point computation and memory bandwidth, making them impractical for low-power devices. Spiking neural networks (SNNs) promise efficiency through sparse, event-driven communication, yet current ANN-to-SNN conversion pipelines still rely on floating-point softmax, RMSNorm and SwiGLU/SiLU or fall back to ReLU-compatible spiking surrogates, often requiring fine-tuning to recover accuracy. This work introduces a family of spike-friendly approximations that collectively replace softmax, RMSNorm and SwiGLU/SiLU. Each operator is built from simple shifts, comparisons and integer additions, requires no lookup tables or floating-point units, and can be seamlessly integrated into existing conversion pipelines without fine-tuning the original weights. We provide theoretical error bounds and integrate the approximations into the SNN conversion pipeline for LLaMA models. Experiments show that the resulting fully spike-driven LLMs maintain performance comparable to SNN large models with floating-point activations, while avoiding the need for post-conversion fine-tuning. These results pave a practical path toward deploying large Transformers on neuromorphic hardware.

## 1 INTRODUCTION

Artificial Neural Networks (ANNs), particularly Transformer-based architectures Vaswani et al. (2017), have driven breakthroughs in vision, speech, and language Krizhevsky et al. (2012); Devlin et al. (2019). Among them, Large Language Models (LLMs) such as GPT-4, LLaMA Touvron et al. (2023); Dubey et al. (2024), and Qwen Hui et al. (2024) have advanced the frontier of natural language understanding and generation. However, their reliance on computationally intensive floating-point operations and large memory footprints presents critical challenges for deployment in edge or low-power environments.

In contrast, Spiking Neural Networks (SNNs) have emerged as a biologically plausible and energy-efficient alternative to ANNs. By leveraging sparse, event-driven communication, SNNs replace costly multiply–accumulate operations with accumulations of binary spike events, substantially reducing energy consumption and latency Roy et al. (2019). This efficiency is particularly appealing for edge inference and neuromorphic platforms such as Intel Loihi Davies et al. (2021). In vision tasks, SNNs have demonstrated competitive performance using spike-based CNNs and ResNet architectures Zheng et al. (2021); Kim & Panda (2021); Rathi & Roy (2020). Building on this foundation, recent works such as SpikeGPT Zhu et al. (2023) and SpikeBERT Lv et al. (2023) have proposed end-to-end spiking language models by introducing spiking self-attention blocks and training with surrogate gradients, demonstrating the feasibility of spike-based language modeling with low-latency generation and moderate accuracy retention.

While directly trained spiking language models have shown early promise, their reliance on large-scale datasets and substantial compute renders them impractical at LLM scale. This has motivated a new direction inspired by the observation that quantized LLMs and spike-driven models share similar arithmetic patterns—both relying on lightweight, discretized operations. This insight has led

to the development of ANN-to-SNN conversion strategies, which map pre-trained ANNs to spiking counterparts by replacing key modules with spike-compatible approximations.

For example, SpikeZIP-TF You et al. (2024) replaces attention, feed-forward, and normalization layers with time-separable, spike-friendly modules, enabling per-timestep parallelism and event-driven execution. Similarly, SorBET Tang et al. (2025) introduces shift-based approximations for softmax and normalization, allowing spiking inference in BERT after modest fine-tuning. These approaches mark promising steps toward scalable, spike-native LLMs. However, existing conversion pipelines still face two fundamental challenges:

- **Activation Mismatch.** Most conversion frameworks hard-code `ReLU`-compatible integrate-and-fire neurons, while modern LLMs widely adopt `SiLU` activations for faster convergence and improved generalization. Prior work has shown that naively replacing these nonlinearities with `ReLU` (or its spiking surrogate) leads to noticeable degradation in language model quality (Chen et al., 2025a). This gap highlights an open problem: designing spike-friendly yet functionally faithful approximations to `SiLU`.

- **Floating-point Residues.** Key components of the Transformer architecture—`softmax` Goodfellow et al. (2016), `RMSNorm` Zhang & Sennrich (2019), and `SiLU` Elfwing et al. (2018); Hendrycks & Gimpel (2016)—continue to rely on floating-point operations such as `exp`, division, or large lookup tables in state-of-the-art ANN-to-SNN conversion pipelines (You et al., 2024; Tang et al., 2025). These *floating-point residues* undermine the promised efficiency of SNNs, introduce additional memory buffers, and complicate deployment on neuromorphic hardware.

Attempts to address these limitations often involve costly fine-tuning or multi-stage distillation (Sanh et al., 2019), which compromises the generality and plug-and-play nature of ANN-to-SNN conversion. These observations expose a central contradiction: while SNNs are valued for low-power, low-memory inference, existing pipelines remain dependent on float-based operations and surrogate activations that erode both efficiency and fidelity. This tension motivates the following question:

**Research Question.** *Can we design a family of SNN-friendly functions that avoid floating-point operations and fine-tuning, thereby enabling low-memory, spike-native, and Transformer-compatible architectures?*

In this work, we take a first step toward answering this question. We introduce a novel set of spike-friendly approximations for softmax, normalization, and activation functions that:

- rely only on simple arithmetic (e.g., bit shifts, comparisons, integer additions),
- are compatible with quantized, low-bit spiking systems,
- preserve functional fidelity to their ANN counterparts even in large-scale Transformer models, and
- require **no additional fine-tuning or retraining**.

Our experiments show that these approximations can be seamlessly integrated into ANN-to-SNN conversion pipelines for Transformer-based models, resulting in fully spike-driven LLMs that preserve strong performance while avoiding pulse-unfriendly computations.

## 2 PRELIMINARY AND RELATED WORK

To support our method, we begin by reviewing essential background on spiking neuron models and the common practices in converting artificial neural networks (ANNs) into spiking neural networks (SNNs). We also revisit two fundamental processing paradigms—Time-Independent and Time-Dependent forms—widely used in recent SNN-based architectures.

### 2.1 SPIKING NEURONS AND SNN DYNAMICS

**Time-Unfolded Computation and Rate Fidelity.** SNNs evolve over discrete timesteps $t = 1, \ldots, T$, representing a temporal processing window $\mathcal{T}$. In the basic Integrate-and-Fire (IF) model,

each neuron emits a binary spike $s^t \in \{0, 1\}$ per timestep, while extended variants allow integer-valued spikes $s^t \in \mathbb{N}$. To approximate the real-valued activation $a_{\text{ANN}}$ of an artificial neuron, these spikes are integrated over time, scaled by a firing threshold $\theta$:

$$a_{\text{SNN}} \approx \sum_{t=1}^{T} s^t \cdot \theta, \quad \mathbb{E}[s^t] \cdot \theta \cdot T \approx a_{\text{ANN}}. \tag{1}$$

Here, $\theta$ denotes the spike amplitude and serves as a quantization scale. In ANN-to-SNN conversion pipelines, this provides a natural foundation: the spike count encodes the quantized magnitude of the original activation, and $\theta$ is matched to the scaling factor in the quantized ANN Rueckauer et al. (2017).

**From IF to IS Neurons.** While binary IF neurons suffice for low-precision tasks, they struggle to represent the rich activations required in modern architectures like Transformers. To address this, recent works adopt *Integrate-and-Shift (IS)* neurons that emit more granular, integer-valued spikes.

A representative IS neuron model is defined as follows. Let $v^l(t)$ denote the membrane potential at layer $l$ and time $t$, and $I^l(t)$ the input current. The neuron updates its internal state and emits a multi-bit spike $s^l(t)$:

$$m^l(t) = v^l(t-1) + \hat{I}^l(t) + I_z^l, \quad v^l(t) = m^l(t) - s^l(t)\theta^l, \tag{2}$$

$$s^l(t) = \begin{cases} L, & m^l(t) \geq L\theta^l, \\ k, & k\theta^l \leq m^l(t) < (k+1)\theta^l, 1 \leq k < L, \\ 0, & \text{otherwise}, \end{cases} \tag{3}$$

$$\hat{I}^l(t) = I^l(t) - I_z^l, \quad \hat{O}^l(t) = s^l(t)\theta^l - I_z^l. \tag{4}$$

Here, $\theta^l$ is the firing threshold, $L$ is the maximum spike level, and $I_z^l$ is the zero-point bias. The output $\hat{O}^l(t)$ serves as a real-valued proxy for ANN activations, making this neuron compatible with components such as LayerNorm and attention mechanisms.

**Remark 1** (Unified View of Multi-Valued Spiking Neurons). *There exist various forms of multi-valued spiking neurons in prior work. For example, integer-spike neurons are widely adopted in models such as SpikeZIP-TF You et al. (2024) and LAS Chen et al. (2025b), while multi-threshold neurons are extensively used in LM-HT SNN Hao et al. (2024). Despite differences in implementation, these designs share a common foundation: enabling multi-bit spike emission while retaining membrane potential, thereby bridging the gap between ANN and SNN representations Tang et al. (2025). As such, they can all be interpreted within a unified framework based on the Integrate-and-Shift (IS) neuron model. Since $k$ is typically a small integer, the term $k\theta^l$ can be implemented using repeated additions instead of multiplications, further reducing hardware cost.*

## 2.2 ANN-TO-SNN CONVERSION PARADIGM

**Conversion-Based Strategy.** A widely adopted strategy for building deep SNNs is to *convert* a pre-trained artificial neural network (ANN)—often already quantized—into a spiking counterpart by replacing key modules with spike-compatible surrogates Cao et al. (2015); Rueckauer et al. (2017). This approach preserves the expressive power of ANNs while avoiding the high training cost of deep SNNs, and has demonstrated strong empirical success across both vision and sequential domains Tang et al. (2025); You et al. (2024); Chen et al. (2025b). To adapt to the discrete and sparse nature of spike trains, prior work has leveraged spiking-specific computation schemes. For linear operations like matrix multiplication, spike-triggered accumulation enables efficient approximations using elementwise operations or other alternatives Zhou et al. (2023), which simplify hardware deployment. In contrast, nonlinear and normalization layers—such as softmax, SiLU, and RMSNorm—pose a more fundamental challenge, as their non-additive nature leads to mismatches when directly applied to temporally evolving spikes. Addressing this barrier is essential for faithful ANN-to-SNN conversion and is the focus of our work.

**Challenges with Nonlinear Functions.** In contrast, nonlinear and normalization layers—such as softmax, SiLU, and RMSNorm—pose a more fundamental challenge. These functions are typically *non-additive*, meaning they violate the identity $\phi(\sum_t \mathbf{x}(t)) \neq \sum_t \phi(\mathbf{x}(t))$, which leads to

Figure 1: Comparison of Time-Independent (TIF) and Time-Dependent (TDF) schemes.

semantic mismatches when operating on temporally distributed spikes. This misalignment not only compromises functional fidelity but also hampers compatibility with neuromorphic hardware Rueckauer et al. (2017). Addressing this issue is therefore a key obstacle for faithful and efficient ANN-to-SNN conversion. To address the non-additivity barrier, recent works have proposed two temporal strategies tailored to spike-based processing Tang et al. (2025); You et al. (2024), as illustrated in Figure 1. We adopt the terms *Time-Independent Form (TIF)* and *Time-Dependent Form (TDF)* to distinguish these schemes.

- **TIF** –integrate spikes over a window, then apply $\phi$ once:

$$\mathbf{X}(t) = \mathbf{X}(t-1) + \mathbf{x}(t), \ \mathbf{o}(t) = \phi\big(\mathbf{X}(t)\big) - \phi\big(\mathbf{X}(t-1)\big), \ \sum_t \mathbf{o}(t) = \phi\Big(\sum_t \mathbf{x}(t)\Big) \tag{5}$$

- **TDF** – apply $\phi$ at each timestep and then sum: $\mathbf{o} = \sum_{t=1}^{T} \phi\big(\mathbf{x}(t)\big)$.

**Limitations and Our Goal.** Although TIF and TDF offer structured work-arounds, they ultimately invoke floating-point exponentials, divisions, or square roots, incurring significant memory and compute overhead Tang et al. (2025); Chen et al. (2025a). These operations are fundamentally incompatible with spike-based hardware, which favors integer, bitwise, or accumulation-only computations for energy and latency efficiency. Building on this line of research, we propose spike-native approximations for these nonlinear operations that eliminate floating-point arithmetic while preserving Transformer fidelity.

## 3 METHOD

In this section, we propose spike-friendly approximations for key components in Transformer-based models, aiming to achieve **spike-friendliness** ANN-to-SNN conversion. These models typically employ SwiGLU activations and RMSNorm normalization, which pose significant challenges due to their nonlinear and division-heavy structures.

### 3.1 EASYSPIKING FRAMEWORK

To address the computational bottlenecks outlined above, we introduce a principled approximation framework called **EasySpiking**, which reformulates core nonlinear operations in Transformer models to be spike-compatible and hardware-efficient. We define a function or operation as *spike-friendly* if it can be implemented using only floating-point additions, subtractions, scalar multiplications (by constants), comparisons, and bit-shifts, while avoiding general-purpose floating-point multiplications, divisions, exponentials, or square roots. This definition reflects the primitive operations supported by modern neuromorphic hardware Davies et al. (2018); Akopyan et al. (2015), which favor linear arithmetic and event-driven execution. However, state-of-the-art Transformer-based models rely on highly nonlinear and floating-point-heavy components. In particular, three functions form the primary barriers to full spike compatibility (see Figure 2):

$$\phi_{\text{Softmax}}(x_i) = \frac{e^{x_i}}{\sum_j e^{x_j}}, \ \phi_{\text{SiLU}}(x) = \frac{x}{1 + e^{-x}}, \ \phi_{\text{RMSNorm}}(x) = \frac{x}{\sqrt{\frac{1}{d}\sum_{i=1}^{d} x_i^2 + \epsilon}} \tag{6}$$

A key observation is that all of these functions can be reformulated as a *fractional structure*, where a nonlinear numerator is divided by a nonlinear or norm-based denominator. Based on this insight, the EasySpiking framework decomposes the approximation into two stages:

Figure 2: **Overview of the EasySpiking framework.** Each non-spike-friendly function (SiLU, Softmax, RMSNorm) is approximated using modular spiking Blocks: the Piecewise Linear Exponential (PWL-EXP) Unit, PolarNorm Unit, and Division Neuron.

1. We design spike-friendly approximations for the numerator and denominator separately for different functions, using Piecewise Linear Exponential Unit and PolarNorm Unit.

2. We introduce a novel **division neuron**, which accepts two spike-based inputs (numerator and denominator) and performs a discretized division over time based on IS neuron.

This modular design enables each complex function to be approximated by chaining primitive spiking units.

**TIF and TDF Compatibility**   As shown in Figure 1, the main difference between TIF and TDF lies in the timing of spike generation. TDF requires an explicit IS neuron to emit the final output, whereas TIF only needs to approximate $\phi$ once and emit spikes per timestep according to Equation equation 5. Our division neuron is built upon the IS neuron mechanism and thus naturally supports TDF-style evaluation. When implementing TIF, we simply configure the division neuron to complete the output within a single timestep. Detailed configurations are provided in the next subsection.

## 3.2 CORE BUILDING BLOCKS

We now present the core building blocks of the EasySpiking framework, starting with the **division neuron**—a central component for realizing the fractional structures described earlier and enabling both Time-Independent and Time-Dependent spiking approximations.

### 3.2.1 DIVISION NEURON

To enable spike-friendly approximations of fractional functions such as Softmax, SiLU, and RMSNorm, we introduce the **division neuron**, which emits spike sequences approximating the ratio $I_A/I_B$ between two spike-coded inputs.

This design extends the Integrate-and-Shift (IS) neuron, which supports multi-level spiking and residual voltage retention. Here, $I_A$ serves as the numerator and is injected into the membrane, while $I_B$ serves as the denominator and defines the threshold level. The neuron emits $L$ discrete

spike levels over $T$ timesteps:

$$\theta^l = I_B \gg n, \quad m^l(t) = v^l(t-1) + \hat{I}_A^l(t) + I_z^l, \tag{7}$$

$$s^l(t) = \begin{cases} L, & m^l(t) \geq L\theta^l, \\ k, & k\theta^l \leq m^l(t) < (k+1)\theta^l, \quad 1 \leq k < L, \\ 0, & \text{otherwise}, \end{cases} \tag{8}$$

$$v^l(t) = m^l(t) - s^l(t)\theta^l, \quad \hat{I}^l(t) = I^l(t) - I_z^l, \quad \hat{O}^l(t) = s^l(t) \gg n - I_z^l, \tag{9}$$

where $n$ is a shift factor, $\hat{I}_A^l(t)$ is the actual injected current and $\gg$ is bit-shifts operator. Both $T$ (timesteps) and $L$ (spike levels) are powers of two.

In the TDF mode, $\hat{I}_A^l(0) = I_A$ and $\hat{I}_A^l(t) = 0$ for $t > 0$. The neuron emits at most $L$ spikes over $T$ steps with threshold $\theta^l = I_B \gg \log_2(TL)$. And the TIF variant compresses the computation into a single timestep, setting $\hat{I}_A^l = I_A$ and $n = L$, allowing immediate emission of all spikes. Both modes support division using only shifts and accumulations.

The choice between TDF and TIF reflects a trade-off between hardware constraints and runtime performance in ES-Framework. We recommend adopting TDF in resource-constrained environments, and TIF in high-performance scenarios where immediate response is required.

### 3.2.2 PIECEWISE LINEAR EXPONENTIAL UNIT (PWL-EXP UNIT)

The core challenge in both SiLU and Softmax lies in the exponential term $\exp(x)$, which is not naturally spike-friendly. To address this, we introduce the PWL-Exp Unit.

We restrict our approximation to the interval $[-L, L]$, which covers the effective input range for most normalized activations. This range is divided into K uniform segments. Let $\gamma = 2L/K$ and we apply piecewise linear interpolation for each subinterval:

$$e^x \approx ax + b = \frac{e^{x_{i+1}} - e^{x_i}}{x_{i+1} - x_i}(x - x_i) + e^{x_i}, \quad x_i = -L + \gamma i, \quad i = 0, 1, \ldots, K-1.$$

The coefficients $a$ and $b$ are precomputed and stored in lookup tables. To minimize memory usage, the coefficient $a$ is quantized to 8-bit fixed-point precision. This design eliminates the need for online exponential computation and enables high-throughput deployment in SNNs.

The PWL-Exp Unit introduces negligible overhead and operates with a cost comparable to a lightweight linear mapping. Its structure is particularly amenable to LUT-based implementations in neuromorphic hardware.

**PolarNorm Unit (PN Unit).** To replace the denominator in RMSNorm (6) with a hardware-friendly approximation, we introduce the *PolarNorm Unit (PN Unit)*. Specifically, we consider the expression $\sqrt{\sum_{i=1}^{d} x_i^2 + \epsilon d}$ and aim to approximate it using only additions, subtractions, comparisons, and bit-shifts—without general-purpose multiplications or square roots.

Let $\mathbf{v} = [x_1, x_2, \ldots, x_d, \sqrt{\epsilon d}]$ be the augmented input vector. Our goal is to estimate $\|\mathbf{v}\|_2$ via a recursive reduction process using the *CORDIC-Hypot* algorithm Volder (1959), which approximates $\sqrt{x^2 + y^2}$ with only shift-add logic.

The approximation is performed using a binary-tree structure: adjacent elements of $\mathbf{v}$ are recursively merged via CORDIC-Hypot operations. Each CORDIC step updates $(x_k, y_k)$ as

$$x_{k+1} = x_k + d_k \cdot \frac{y_k}{2^k}, \quad y_{k+1} = y_k - d_k \cdot \frac{x_k}{2^k}, \tag{10}$$

where $d_k = \text{sign}(y_k)$. After $n$ iterations, the output $x_n$ approximates $\sqrt{x^2 + y^2}$ up to a constant gain. Applying this recursively over the tree yields a scalar norm estimate. The final result is rescaled by a fixed inverse gain $1/K_n$ to correct for CORDIC accumulation. This constant factor does not violate spike-friendly constraints: it can be absorbed into the scale $theta$ or approximated using fixed-point scaling.

### 3.3 EASYSPIKING FUNCTION (ES-FUNCTION)

We now build spike-friendly versions of key nonlinear functions using the previously introduced modules: the Division Neuron, the PWL-Exp Unit, and the PolarNorm Unit. Each function is decomposed into a numerator and a denominator, approximated independently, and combined via the Division Neuron.

**EasySpiking-Softmax (ES-Softmax).** We first stabilize the input using the shift-invariance property of Softmax:

$$z_i \leftarrow z_i - \max_j z_j + L,$$

ensuring $z_i \in (-\infty, L]$. Values below $-L$ are clamped to zero to suppress negligible contributions.

- **Numerator:** $\exp(z_i)$ is approximated using the PWL-Exp Unit within the interval $[-L, L]$.
- **Denominator:** The summation $\sum_j \exp(z_j)$ is computed through temporal accumulation over $T$ timesteps.
- **Output:** The final normalized output is obtained via the Division Neuron.

This formulation avoids runtime exponentiation and division, enabling low-latency spike-based inference with bounded output.

**EasySpiking-SiLU (ES-SiLU).** For SiLU, we constrain the domain to $[-L, L]$ and extend the output linearly beyond this range: $\text{SiLU}(x) \leftarrow x$ for $x > L$, and $\text{SiLU}(x) \leftarrow 0$ for $x < -L$.

- **Numerator:** The input $x$ is directly encoded as a spike train.
- **Denominator:** The term $1 + \exp(-x)$ is approximated by applying the PWL-Exp Unit to $-x$ and adding 1.
- **Output:** The final normalized value is computed via the Division Neuron.

This construction preserves the smooth nonlinearity of SiLU while maintaining spike compatibility.

**EasySpiking-RMSNorm (ES-RMSNorm).** To approximate RMSNorm, we follow the transformed formulation:

- **Numerator:** Each $x_i$ is scaled by the constant $\sqrt{d}$ and encoded as a spike train.
- **Denominator:** The expression $\sqrt{\sum x_i^2 + \epsilon d}$ is approximated using the PolarNorm Unit.
- **Output:** The final normalized value is computed via the Division Neuron.

All EasySpiking functions adhere to the same modular construction pattern, relying exclusively on primitive spike-friendly operations. This enables seamless deployment on low-power neuromorphic systems without sacrificing functional fidelity, as evaluated in the following Section.

## 4 PERFORMANCE ANALYSIS

To validate the effectiveness of the EasySpiking framework, we analyze its performance from two complementary perspectives: *accuracy* and *memory footprint*, aiming to provide a holistic view of how spike-based approximations balance functional fidelity with hardware constraints. Throughout this section we denote by:

$$\varepsilon_{\exp} = \frac{L^2}{2K^2} e^{2L/K}, \quad \Delta = \frac{1}{n}, \quad \varepsilon_{\text{pol}} = \lceil \log_2 d \rceil \, 2^{-2n-1},$$

the relative error of PWL-Exp, the quantisation step of the $(T, L)$-Division neuron, and the relative $\ell_2$–norm error of an $n$–step CORDIC tree in the PolarNorm Unit, respectively.

**Theorem 1** (Error Bounds for EasySpiking). *For each EasySpiking function $\tilde{\phi}$, assume the standard spike-based approximations (PWL-Exp, Division Neuron, and PolarNorm) are employed. Then the following per-output error bounds hold:*

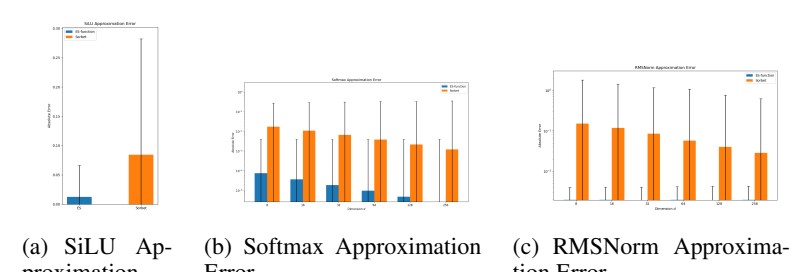

(a) SiLU Approximation Error

(b) Softmax Approximation Error

(c) RMSNorm Approximation Error

(d) TDF and TIF Error

Figure 3: Function-level and module-level approximation error comparisons between **ES-function** and **Sorbet**. (a)–(c) report function-level absolute errors for SiLU, Softmax, and RMSNorm, respectively, all evaluated under 8-bit quantization. (d) shows the mean $L_2$ error between SNN and ANN outputs for different conversion strategies on 1,000 MMLU samples, under varying time steps $T$.

> *(i)* ***ES-Softmax.*** *For every class $i$,* $\frac{|\tilde{\phi}_i - \phi_i|}{\phi_i} \leq \frac{2}{1 - \varepsilon_{\exp}} \left( \varepsilon_{\exp} + \Delta \right)$.

> *(ii)* ***ES-SiLU.*** *For every $x \in [-L, L]$,* $|\tilde{\phi}(x) - \phi(x)| \leq |x| \frac{2\varepsilon_{\exp}}{1 - \varepsilon_{\exp}} + |x| \Delta$.

> *(iii)* ***ES-RMSNorm.*** *For each coordinate $i$,* $\frac{|\tilde{\phi}_i - \phi_i|}{\phi_i} \leq \frac{\varepsilon_{pol} + \Delta}{1 - \varepsilon_{pol}} + \sqrt{d} \, \Delta$.

These bounds demonstrate that EasySpiking functions achieve high-precision approximations across core nonlinearities. Each error term remains tightly controlled, stemming from well-isolated sources: lookup-based exponentiation, spike-based division, and recursive norm estimation. We empirically validate these guarantees in the next section. The proof is available in the Appendix.

**Memory Efficiency.** Compared to traditional floating-point computation, EasySpiking only requires storing a compact set of lookup entries: $K$ values of 8-bit and 16-bit precision in total. This is significantly smaller than typical table-based methods, which often rely on large floating-point tables. Such reduction is critical for deployment on spiking neuromorphic chips, which usually operate under strict on-chip memory constraints.

**Remark 2** (Recommended Setting). *From the preceding discussion, it follows that there is a trade-off between accuracy and memory usage. To ensure sufficient fidelity, we require $\varepsilon_{\exp} < 10^{-2}$. Thus we recommend $L = 5, K = 64$, where $\varepsilon_{\exp} \leq 3.63 \times 10^{-3}$ and the memory footprint remains acceptable.*

## 5 EXPERIMENTS

To evaluate the proposed ES-Framework, we conduct experiments across three levels: (1) function level, (2) module level, and (3) model level. These levels reflect a progressively broader view of how our spike-friendly design impacts accuracy, efficiency, and deployability. Based on this hierarchy, we design four experiments that jointly assess the functional fidelity, hardware cost, and end-to-end performance of EasySpiking. In addition, we provide more comparative results in Appendix B and detailed ablation studies in Appendix C to further verify the reliability of our algorithm. All experiments are executed on four NVIDIA H20 GPUs in parallel.

**Function-Level Analysis.** We begin by evaluating our spike-friendly approximations of key nonlinear functions, including $\phi_{\text{Softmax}}$, $\phi_{\text{SiLU}}$, and $\phi_{\text{RMS}}$. We report their maximum absolute error, mean relative error, and hardware cost under integer-only implementations. Results show that our approximations achieve high fidelity while remaining fully compatible with bitwise operations.

As shown in Figure 3(a-c), the ES-function consistently achieves lower approximation errors compared to Sorbet across all three functions. For $\phi_{\text{SiLU}}$, the ES variant reduces the mean absolute error to less than 0.015, while Sorbet exhibits significantly higher variability and peak error over the extended test domain. For $\phi_{\text{Softmax}}$, ES maintains stable accuracy across all input dimensions, achieving mean errors below $10^{-3}$, whereas Sorbet's error remains above $10^{-2}$ and increases with

Table 1: Performance on LLaMA Models. We report *acc* for WinoGrande and *acc_norm* for HellaSwag, ArcE, and PIQA.

| Model | Method ($T$) | Precision | WinoGrande | HellaSwag | ArcC | ArcE | PIQA | Avg. Acc. |
|-------|--------------|-----------|------------|-----------|------|------|------|-----------|
| 2-7B | SNN ($T=1$) | W6A6/W8A8 | 70.09 / 70.24 | 74.06 / 76.44 | 44.80 / 45.99 | 73.11 / 73.02 | 77.15 / 78.35 | 67.84 / 68.81 |
| | ES ($T=1$) | W6A6/W8A8 | 67.48 / 68.35 | 73.87 / 76.23 | 44.71 / 46.50 | 73.32 / 73.86 | 76.44 / 78.62 | 67.16/ 68.71 |
| | SNN ($T=2$) | W6A6/W8A8 | 69.06 / 69.93 | 74.23 / 76.45 | 44.88 / 45.90 | 72.98 / 72.90 | 75.68 / 78.45 | 67.37 / 68.73 |
| | ES ($T=2$) | W6A6/W8A8 | 69.53 / 69.61 | 74.17 / 76.46 | 44.45 / 46.33 | 73.06 / 73.11 | 76.77 / 78.35 | 67.60 / 68.77 |
| | SNN ($T=4$) | W6A6/W8A8 | 69.46 / 69.85 | 74.11 / 76.59 | 45.14 / 46.16 | 72.98 / 73.40 | 76.82 / 78.24 | 67.70 / 68.85 |
| | ES ($T=4$) | W6A6/W8A8 | 68.51 / 69.30 | 73.11 / 76.34 | 43.86 / 45.90 | 72.73 / 74.20 | 77.09 / 78.40 | 67.06 / 68.83 |
| 3-8B | SNN ($T=1$) | W6A6/W8A8 | 71.82 / 73.09 | 77.61 / 78.96 | 50.94 / 53.75 | 75.59 / 77.99 | 77.69 / 80.47 | 70.73 / 72.85 |
| | ES ($T=1$) | W6A6/W8A8 | 74.11 / 73.72 | 77.40 / 79.04 | 49.40 / 53.41 | 75.59 / 77.65 | 77.75 / 80.36 | 70.85 / 72.84 |
| | SNN ($T=2$) | W6A6/W8A8 | 71.82 / 73.16 | 77.75 / 79.01 | 47.27 / 53.75 | 75.21 / 77.86 | 75.84 / 79.98 | 69.58 / 72.75 |
| | ES ($T=2$) | W6A6/W8A8 | 72.22 / 73.24 | 77.58 / 78.83 | 48.89 / 53.50 | 75.63 / 77.57 | 77.86 / 80.03 | 70.44 / 72.63 |
| | SNN ($T=4$) | W6A6/W8A8 | 70.40 / 73.32 | 77.65 / 78.91 | 48.98 / 53.58 | 74.33 / 80.43 | 75.90 / 79.98 | 69.45 / 73.24 |
| | ES ($T=4$) | W6A6/W8A8 | 70.24 / 73.01 | 77.31 / 79.03 | 48.55 / 52.56 | 75.34 / 78.20 | 76.01 / 80.20 | 69.49 / 72.60 |
| 3-70B | SNN ($T=1$) | W8A8 | 79.32 | 85.65 | 62.37 | 82.79 | 84.11 | 78.85 |
| | ES ($T=1$) | W8A8 | 78.85 | 85.71 | 62.54 | 82.20 | 83.90 | 78.64 |
| | SNN ($T=2$) | W8A8 | 79.48 | 85.70 | 62.88 | 82.87 | 83.90 | 78.97 |
| | ES ($T=2$) | W8A8 | 79.08 | 85.60 | 62.88 | 82.62 | 83.90 | 78.82 |

dimensionality. In the case of $\phi_{\text{RMS}}$, the ES-function achieves nearly two orders of magnitude lower error than Sorbet, demonstrating strong numerical stability under quantized evaluation.

Although Sorbet offers an alternative approximation strategy, it relies on dedicated fine-tuning and is limited to BERT-style architectures. Moreover, its design philosophy differs fundamentally from our zero-shot ANN-to-SNN conversion approach. Therefore, we exclude Sorbet from subsequent evaluations at both the module and model levels.

**Module-Level Evaluation.** We take LLaMA3-QCFS (with 8-bit weights and activations, 8B scale) as the base model and evaluate three different conversion strategies: (1) a direct ANN-to-SNN conversion without approximation, (2) conversion using the ES framework in time-independent form (ES), and (3) conversion using the time-dependent form (ES-TDF). For evaluation, we randomly sample 1,000 tasks from the MMLU dataset and run inference using each of the three converted models to assess their module-level performance under different design configurations.

Figure 3(d) presents the mean $L_2$ error on 1,000 MMLU samples across different time steps $T \in \{1, 2, 4, 8\}$. The results show that both ES and ES-TDF maintain approximation errors comparable to the direct ANN-to-SNN conversion baseline. Notably, as $T$ increases, the gap among all three configurations narrows, suggesting that spike accumulation compensates for early approximation gaps. These results validate the functional fidelity of our ES-based modules under typical SNN execution settings.

**Model-Level Evaluation.** To assess the scalability and effectiveness of our ES framework at the full model level, we conduct evaluations on LLaMA2-7B, LLaMA3-8B, and LLaMA3-70B. Each model is first converted into spiking form via ANN-to-SNN conversion under two quantization settings: W6A6 and W8A8. For each setting, we compare three conversion variants: a baseline SNN without approximation, ES-TIF, and ES-TDF. All evaluations are performed at time steps $T \in \{1, 2, 4\}$ to capture both low-latency and high-accuracy behaviors.

Table 1 summarizes the end-to-end performance of our ES-converted models on five representative language understanding tasks: Winogrande, HellaSwag, ARC-Challenge, ARC-Easy, and PIQA. We evaluate both standard SNN baselines and ES-based conversions on LLaMA2-7B and LLaMA3-8B under W6A6 and W8A8 quantization settings, across time steps $T \in \{1, 2, 4\}$, and on LLaMA3-70B under W8A8 across time steps $T \in \{1, 2\}$.

All ES results in this table correspond to the time-independent form (ES-TIF). As shown, ES-TIF models achieve accuracy comparable to direct SNN conversions across most benchmarks. At low time steps ($T = 1$), ES-TIF retains competitive accuracy while remaining fully compatible with integer-only execution. As $T$ increases, the performance gap among methods narrows, confirming the stability of ES approximations under deeper spike integration. Due to space limitations, additional results—such as ES-TDF performance are provided in the appendix.

ETHICS STATEMENT

This work adheres to the ICLR Code of Ethics. In this study, no human subjects or animal experimentation was involved. All datasets used were sourced in compliance with relevant usage guidelines, ensuring no violation of privacy. We have taken care to avoid any biases or discriminatory outcomes in our research process. No personally identifiable information was used, and no experiments were conducted that could raise privacy or security concerns. We are committed to maintaining transparency and integrity throughout the research process.

REPRODUCIBILITY STATEMENT

We have made every effort to ensure that the results presented in this paper are reproducible. All code is included in the supplementary material, together with a complete README and launch scripts that enable one-click reproduction of all experiments. The experimental setup, including training steps, model configurations, and hardware details, is described in detail in the paper. Additionally, all datasets used in this work are publicly available, ensuring consistent and reproducible evaluation results. We believe these measures will enable other researchers to reproduce our work and further advance the field.

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

# A    PROOF OF THEOREM 1

Based on the method introduced in the previous section, we have derived a set of spike-compatible ES-functions intended for application in the forward propagation of the spike-LLM. We now estimate the approximation errors of the three ES-functions used in this work: ES-Softmax, ES-SiLU, and ES-RMSNorm. First, we need some necessary lemmas.

**Lemma 1** (PWL-Exp Unit Relative Error Bound). *Let $\tilde{e}^x$ be the piecewise-linear approximation to $e^x$ on $[-L, L]$ obtained by dividing the interval into $K$ equal subintervals of width*

$$h = \frac{2L}{K},$$

*and interpolating linearly on each. Then for every $x \in [-L, L]$,*

$$\left| \frac{\tilde{e}^x - e^x}{e^x} \right| \leq \frac{h^2}{8} e^h = \frac{\left(\frac{2L}{K}\right)^2}{8} e^{2L/K}.$$

*Especially, when $L = 5, K = 64$, we have:*

$$\left| \frac{\tilde{e}^x - e^x}{e^x} \right| \leq \frac{h^2}{8} e^h = \frac{\left(\frac{5}{32}\right)^2}{8} e^{5/32} \approx 3.63 \times 10^{-3}.$$

**Lemma 2.** *Volder (1959)[Pairwise CORDIC Relative Error Bound] Let $a, b \in \mathbb{R}$ and $r = \sqrt{a^2 + b^2}$. Perform $n$ steps of CORDIC in vectoring mode with angles $\phi_i = \arctan(2^{-i})$ for $i = 0, 1, \ldots, n - 1$, and apply the exact scale correction*

$$K_n^{-1} = \prod_{i=0}^{n-1} \frac{1}{\sqrt{1 + 2^{-2i}}}.$$

*If $\tilde{r}$ denotes the CORDIC output after correction, then*

$$\frac{|\tilde{r} - r|}{r} \leq 2^{-2n-1}.$$

**Lemma 3** (Binary-Tree CORDIC Accumulation Error). *To compute*

$$R = \sqrt{\sum_{j=1}^{D} v_j^2 + \varepsilon D},$$

*group the $D$ values into $\lceil D/2 \rceil$ pairs, apply Lemma 2 to each pair (using the same $n$-step CORDIC), and then recursively merge the resulting radii in a balanced binary tree of height $\ell = \lceil \log_2 D \rceil$. Denote the final CORDIC result by $\tilde{R}$. Then*

$$\frac{|\tilde{R} - R|}{R} \leq \ell \cdot 2^{-2n-1}.$$

After discussing the error bounds of PWL-Exp Unit and PolarNorm Unit, now we can focus on the errors of ES-Softmax, ES-SiLU, and ES-RMSNorm.

*Proof of Lemma 3.* Let the pairwise CORDIC relative error bound be $\delta := 2^{-2n-1}$. We analyze the error propagation layer by layer through the CORDIC binary tree.

**Base layer (Layer 1):** Group the $D$ inputs into $\lceil D/2 \rceil$ pairs. Apply Lemma 2 to each pair, yielding approximate radii $\tilde{r}_k^{(1)}$ with relative error $\leq \delta$.

**Inductive step (Layer $t$):** Assume inputs to layer $t$ have maximum relative error $\leq (t - 1)\delta$. Merging two such values gives:

$$\tilde{r}^{(t)} = \sqrt{(\tilde{r}_1^{(t-1)})^2 + (\tilde{r}_2^{(t-1)})^2} \cdot (1 + e_{\text{CORDIC}}),$$

where $|e_{\text{CORDIC}}| \leq \delta$ (by Lemma 2).

Each $\tilde{r}^{(t-1)}$ differs from its true value $r^{(t-1)}$ by at most $(t-1)\delta$, so the total error in computing $\tilde{r}^{(t)}$ is at most:

$$|e^{(t)}| \leq (t-1)\delta + \delta = t\delta.$$

**Final result.** The CORDIC reduction forms a complete binary tree whose leaves are the $D$ original vectors. At every layer the number of active nodes is at most halved (merging each pair into one). After $t$ layers the node count is therefore at most $D/2^t$. We need enough layers so that only one node remains:

$$\frac{D}{2^\ell} \leq 1 \quad \Longrightarrow \quad \ell \geq \log_2 D.$$

Taking the smallest integer that meets the inequality yields

$$\ell = \lceil \log_2 D \rceil.$$

Consequently, after $\ell$ layers the relative error of the final radius satisfies

$$\frac{|\tilde{R} - R|}{R} \leq \ell\delta = \ell\, 2^{-2n-1}.$$

$\square$

**Theorem 2** (Error Bound of ES-Softmax). *Let* $\mathbf{z} = (z_1, \ldots, z_d)$ *be the shifted and clipped logits*

$$-L \leq z_i \leq L, \qquad i = 1, \ldots, K,$$

*and write*

$$p_i = \frac{e^{z_i}}{\sum_{j=1}^d e^{z_j}} \quad \text{for the exact softmax, and} \quad \tilde{p}_i = \frac{\tilde{e}^{z_i}}{\sum_{j=1}^d \tilde{e}^{z_j}} + \delta_i$$

*for the ES-Softmax output, where*

*\* $\tilde{e}^{\cdot}$ is the PWL-Exp approximation of Lemma 1 whose relative error satisfies $|\tilde{e}^x - e^x| \leq \varepsilon_{\exp} e^x$ with $\varepsilon_{\exp} = \frac{\left(\frac{2L}{K}\right)^2}{8} e^{2L/K}$;*

*\* the final reciprocal is produced by a $(T, L)$-Division neuron whose quantisation step is $\Delta = \frac{1}{TL}$, so that $|\delta_i| \leq \Delta p_i$ for every $i$.*

*Then, for every class $i$,*

$$\boxed{\frac{|\tilde{p}_i - p_i|}{p_i} \leq \frac{2}{1 - \varepsilon_{\exp}}(\varepsilon_{\exp} + \Delta).}$$

*Proof.* **Step 1: bound the exponential approximations.** By Lemma 1,

$$(1 - \varepsilon_{\exp})e^{z_i} \leq \tilde{e}^{z_i} \leq (1 + \varepsilon_{\exp})e^{z_i}, \qquad i = 1, \ldots, d. \tag{11}$$

**Step 2: bound the denominator.** Summing equation 11 yields

$$(1 - \varepsilon_{\exp})S \leq \tilde{S} := \sum_j \tilde{e}^{z_j} \leq (1 + \varepsilon_{\exp})S, \qquad S = \sum_j e^{z_j}.$$

**Step 3: ratio perturbation.** Define the "ideal" ES-Softmax without quantisation, $\hat{p}_i = \tilde{e}^{z_i}/\tilde{S}$. Using the standard perturbation inequality[1] and equation 11,

$$\frac{|\hat{p}_i - p_i|}{p_i} \leq \frac{2\varepsilon_{\exp}}{1 - \varepsilon_{\exp}}.$$

---

[1]For positive $a, b$ and errors $|\delta a| \leq \varepsilon a$, $|\delta b| \leq \varepsilon b$, one has $\left|(a + \delta a)/(b + \delta b) - a/b\right| \leq 2\varepsilon\, a/b\,(1 - \varepsilon)^{-1}$.

**Step 4: add the division-neuron quantisation.** The Division neuron introduces an extra additive error $|\delta_i| \leq \Delta$, so

$$|\tilde{p}_i - p_i| \ \leq \ |\tilde{p}_i - \hat{p}_i| + |\hat{p}_i - p_i| \ \leq \ (\Delta + \frac{2\varepsilon_{\exp}}{1 - \varepsilon_{\exp}}) \, p_i.$$

Dividing both sides by $p_i$ proves the claimed bound. $\qquad\square$

**Theorem 3** (Error Bound of ES-SiLU). *Assume the input is clipped to $x \in [-5, 5]$ before ES-SiLU is applied. Let*

$$f(x) = \mathrm{SiLU}(x) = \frac{x}{1 + e^{-x}}, \qquad \tilde{f}(x) = x \, \tilde{\sigma}(x) + \delta_{mul},$$

*where*

\* $\tilde{\sigma}(x) = \dfrac{1}{1 + \tilde{e}^{-x}} + \delta_{div}$ *is obtained from the PWL-Exp approximation $\tilde{e}^{\cdot}$ of Lemma 1 (relative error $\varepsilon_{\exp} = 3.63 \times 10^{-3}$) followed by a $(T, L)$-Division neuron whose quantisation step is $\Delta = 1/(TL)$;*

\* $\delta_{div}$ *and $\delta_{mul}$ are, respectively, the additive errors introduced by the Division neuron and by the spike–time multiplication, both satisfying $|\delta_{div}|, |\delta_{mul}| \leq \Delta$.*

*Then, for every $x \in [-5, 5]$,*

$$\boxed{\left| \tilde{f}(x) - f(x) \right| \ \leq \ |x| \frac{2\varepsilon_{\exp}}{1 - \varepsilon_{\exp}} \ + \ |x| \, \Delta}$$

*and in particular, with $(T, L) = (16, 256)$ so that $\Delta = 2^{-12} \approx 2.44 \times 10^{-4}$,*

$$\left| \tilde{f}(x) - f(x) \right| \ \leq \ 0.038 \qquad \textit{for all } x \in [-5, 5].$$

*Proof.* **1. Bounding the sigmoid approximation.** Because $\sigma(x) = 1/(1 + e^{-x})$ is Lipschitz-continuous on $[-5, 5]$ with Lipschitz constant at most $1/4$, the PWL-Exp error of Lemma 1 implies

$$\left| \hat{\sigma}(x) - \sigma(x) \right| = \left| \frac{1}{1 + \tilde{e}^{-x}} - \frac{1}{1 + e^{-x}} \right| \leq \frac{2\varepsilon_{\exp}}{1 - \varepsilon_{\exp}} \, \sigma(x),$$

where $\hat{\sigma}(x) = 1/(1 + \tilde{e}^{-x})$ is the *ideal* reciprocal without quantisation.

**2. Adding division-neuron quantisation.** The Division neuron produces $\tilde{\sigma}(x) = \hat{\sigma}(x) + \delta_{\mathrm{div}}$ with $|\delta_{\mathrm{div}}| \leq \Delta$, hence

$$\left| \tilde{\sigma}(x) - \sigma(x) \right| \leq \frac{2\varepsilon_{\exp}}{1 - \varepsilon_{\exp}} \, \sigma(x) + \Delta. \tag{12}$$

Therefore, using equation 12 ,

$$\left| \tilde{f}(x) - f(x) \right| \leq |x| \frac{2\varepsilon_{\exp}}{1 - \varepsilon_{\exp}} + |x| \, \Delta,$$

which is the claimed bound. For $(T, L) = (256, 16)$ we substitute $\varepsilon_{\exp} = 3.63 \times 10^{-3}$ and $\Delta \approx 2.44 \times 10^{-4}$ to obtain the numerical value. $\qquad\square$

**Theorem 4** (Error Bound of ES-RMSNorm). *Let $\mathbf{x} = (x_1, \ldots, x_d) \in \mathbb{R}^d$ and define*

$$R = \sqrt{\frac{1}{d} \sum_{j=1}^{d} x_j^2 + \varepsilon}, \qquad y_i = \frac{x_i}{R}, \ \ i = 1, \ldots, d.$$

*Denote by $\tilde{R}$ the output of the **PolarNorm** unit that uses an $n$–step CORDIC in a balanced binary tree of height $\ell = \lceil \log_2 d \rceil$ and let*

$$\varepsilon_{pol} := \ell \, 2^{-2n-1} \quad \textit{so that} \quad \left| \frac{\tilde{R} - R}{R} \right| \leq \varepsilon_{pol} \quad \textit{(Lemma 3)}.$$

*Next, let the $(T, L)$–Division neuron produce*

$$\tilde{q} = \frac{1}{\tilde{R}} + \delta_{div}$$

*with quantisation step*

$$\Delta := \frac{1}{TL} \quad (or \ |\delta_{div}|_{\max}),$$

*and finally obtain the ES-RMSNorm output*

$$\tilde{y}_i = x_i \tilde{q} + \delta_{mul}, \qquad |\delta_{mul}| \leq \Delta.$$

*Then, for each coordinate $i = 1, \ldots, d$, we have*

$$\boxed{\frac{|\tilde{y}_i - y_i|}{|y_i|} \leq \frac{\varepsilon_{pol} + \Delta}{1 - \varepsilon_{pol}} + \frac{\Delta}{|x_i|} R}$$

*and, whenever $|x_i| \geq \sqrt{\varepsilon}$ (the usual case in practice), the last term satisfies*

$$\frac{\Delta}{|x_i|} R \leq \sqrt{d} \cdot \Delta.$$

*Proof.* We decompose the total error as

$$\tilde{y}_i - y_i = x_i \big( \tilde{q} - \frac{1}{R} \big) = x_i \Big( \frac{1}{\tilde{R}} - \frac{1}{R} \Big) + x_i \delta_{\mathrm{div}}.$$

**(i) Reciprocal of $\tilde{R}$:** Using the approximation

$$\tilde{R} = R(1 + \eta), \quad |\eta| \leq \varepsilon_{\mathrm{pol}},$$

we have the following identity:

$$\left| \frac{1}{\tilde{R}} - \frac{1}{R} \right| = \frac{|\eta|}{1 + \eta} \cdot \frac{1}{R} \leq \frac{\varepsilon_{\mathrm{pol}}}{1 - \varepsilon_{\mathrm{pol}}} \cdot \frac{1}{R}.$$

This result follows from the first-order approximation and ensures that the error is controlled by the CORDIC approximation precision.

**(ii) Division-neuron quantisation:** Since the quantisation error is bounded by

$$|x_i \delta_{\mathrm{div}}| \leq |x_i| \Delta,$$

we can combine these terms to get:

$$|\tilde{y}_i - y_i| \leq |x_i| \cdot \frac{\varepsilon_{\mathrm{pol}}}{1 - \varepsilon_{\mathrm{pol}}} \cdot \frac{1}{R} + |x_i| \Delta.$$

Dividing both sides by $|y_i| = \frac{|x_i|}{R}$ gives:

$$\frac{|\tilde{y}_i - y_i|}{|y_i|} \leq \frac{\varepsilon_{\mathrm{pol}}}{1 - \varepsilon_{\mathrm{pol}}} + \frac{\Delta}{|x_i|} R.$$

Thus, we obtain the final error bound:

$$\frac{|\tilde{y}_i - y_i|}{|y_i|} \leq \frac{\varepsilon_{\mathrm{pol}} + \Delta}{1 - \varepsilon_{\mathrm{pol}}} + \frac{\Delta}{|x_i|} R.$$

Finally, when $|x_i|$ is sufficiently large (i.e., $|x_i| \geq \sqrt{\varepsilon}$), we can bound the term:

$$\frac{\Delta}{|x_i|} R \leq \sqrt{d} \cdot \Delta,$$

because $R$ is bounded by a factor of $\sqrt{d}$, as derived from the sum over all $x_j$. $\qquad\square$

## B  MORE EXPERIMENTS

### B.1  ADDITIONAL FUNCTION COMPARISONS

In addition to the Sorbet baseline considered in the main text, we compare our spike-friendly approximations against three families of operator-level methods widely adopted in **the quantized neural network** and **efficient ANN** literature:

- **Uniform Quantization (DoReFa-Net)** (Zhou et al., 2016): outputs are uniformly quantized to $k$-bit fixed-point values. This represents the canonical training-free quantization baseline.
- **Hard Thresholding (XNOR-Net)** (Rastegari et al., 2016): nonlinear outputs are binarized via a hard step function, producing extreme low-cost approximations.
- **Polynomial / PWL Expansions** (Higham, 2005): classical numerical analysis methods approximate $\exp(\cdot)$ and related functions using truncated polynomials or piecewise-linear interpolation. We adopt a Pade [2/2] approximation and a 64-segment PWL implementation on $[-8, 8]$.

We emphasize that all these methods are training-free and evaluated purely at the operator level, ensuring a fair comparison with our ES-functions.

**SiLU Approximation.**  Figure 4 reports operator-level errors of different SiLU approximations evaluated on $x \in [-5, 5]$ under 8-bit quantization. We compare ES-SiLU with Sorbet (ReLU surrogate), DoReFa-4b uniform quantization, XNOR-style binarization, a 64-segment PWL-sigmoid, and the hard-swish activation from MobileNetV3. Thanks to the dedicated handling of saturation/edge values, ES-SiLU exhibits *maximum errors* comparable to the 64-segment PWL baseline, but achieves substantially *lower mean errors*, indicating a more concentrated and stable error distribution. In contrast, Sorbet and hard-swish incur significantly larger deviations, while DoReFa-4b and XNOR baselines suffer from high peak and mean errors. Moreover, since our deployment shares a single lookup table across Softmax, SiLU, and RMSNorm, ES-SiLU attains accuracy comparable to strong PWL baselines at an *effective storage cost* close to that of independent PWL-16, maintaining excellent hardware efficiency.

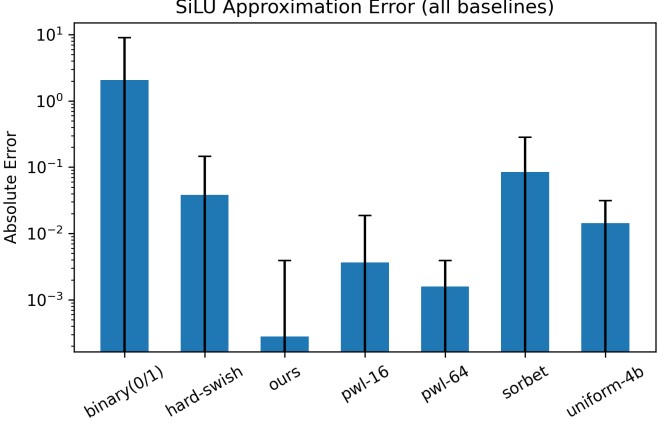

Figure 4: Operator-level errors for SiLU approximations under 8-bit quantization. Error bars indicate the gap between mean and maximum absolute error. Our ES-SiLU consistently achieves the lowest mean error among training-free approximations with low storage cost.

**Softmax Approximation.**  Figure 5 shows operator-level errors for $\phi_{\text{Softmax}}$ across varying input dimensions $d$. We compare ES-Softmax with Sorbet, hardmax, a Pade [2/2] rational approximation, and a 16-segment PWL-exp baseline. Our ES-function consistently achieves the lowest mean error across all dimensions, with maximum error bounded by the 8-bit quantization grid. In contrast,

polynomial and PWL approximations occasionally reach competitive mean accuracy, but incur significantly larger maximum errors and variance, especially at higher dimensions. Sorbet and hardmax baselines exhibit much higher overall error levels. These results demonstrate that ES-Softmax provides both stable accuracy and robustness across dimensions, while maintaining hardware-friendly integer-only implementation.

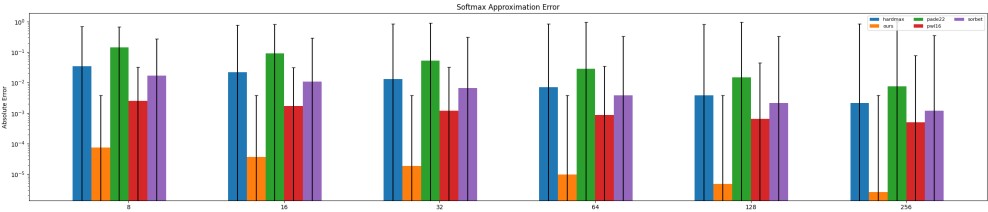

Figure 5: Operator-level errors for Softmax approximations under 8-bit quantization. Error bars indicate the gap between mean and maximum absolute error. Our ES-Softmax achieves the lowest mean error across dimensions while maintaining bounded maximum error under integer-only implementation.

**RMSNorm Approximation.** Figure 6 reports operator-level errors for $\phi_{\mathrm{RMS}}$ across hidden dimensions $d$. We compare our ES-RMS with blockwise RMS baselines (block sizes of 32 and 64), which are commonly adopted in efficient Transformer implementations. Interestingly, blockwise methods occasionally achieve very low error at specific dimensions (e.g., when $d$ is an integer multiple of the block size), since the block partitioning coincides with the full reduction and the normalization becomes exact. However, at other dimensions the same methods incur large deviations due to incomplete aggregation across blocks, leading to unstable performance. By contrast, our ES-RMS maintains consistently low mean error across all tested dimensions, providing stable accuracy under quantized evaluation without relying on dimension-specific coincidences.

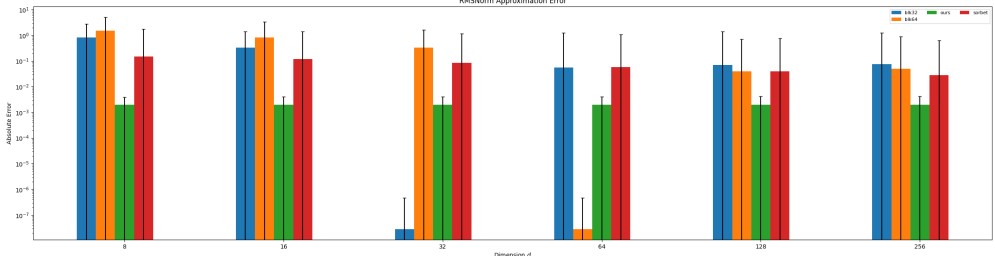

Figure 6: Operator-level errors for RMSNorm approximations under 8-bit quantization. Error bars indicate the gap between mean and maximum absolute error. Our ES-RMS consistently yields lower mean errors than blockwise and Sorbet baselines, and maintains stable performance across dimensions.

### B.2 Additional Model Comparisons

In this section, we mainly present the complete experimental results in model level evaluation. The results include the results of four models, namely the Quantization Scheme InitialQ, DuQ, SNN without ES conversion, SNN converted from ES-TIF, and SNN converted from ES-TDF, on five datasets of WinoGrande, HellaSwag, ArcC, ArcE, PiQA.

Table 2: Performance on LLaMA Models. We report *acc* for WinoGrande and *acc_norm* for HellaSwag, ArcE, and PIQA.

| Model | Method | T | Precision | WinoGrande | HellaSwag | ArcC | ArcE | PIQA | Avg. Acc. |
|---|---|---|---|---|---|---|---|---|---|
| **Llama 2 7B** | PrefixQ | - | W6A6/W8A8 | 70.09 / 70.24 | 74.06 / 76.44 | 44.80 / 45.99 | 73.11 / 73.02 | 77.15 / 78.35 | 67.84 / 68.81 |
| | DuQ | - | W6A6/W8A8 | 67.88 / 66.69 | 72.64 / 72.81 | 40.53 / 40.36 | 53.07 / 53.37 | 77.15 / 77.20 | 62.25 / 62.09 |
| | SNN | 1 | W6A6/W8A8 | 70.09 / 70.24 | 74.06 / 76.44 | 44.80 / 45.99 | 73.11 / 73.02 | 77.15 / 78.35 | 67.84 / 68.81 |
| | **ES-TIF** | 1 | W6A6/W8A8 | 67.48 / 68.35 | 73.87 / 76.23 | 44.71 / 46.50 | 73.32 / 73.86 | 76.44 / 78.62 | 67.16 / 68.71 |
| | **ES-TDF** | 1 | W6A6/W8A8 | 67.64 / 69.22 | 73.73 / 76.34 | 44.71 / 45.99 | 72.81 / 74.03 | 76.99 / 78.51 | 67.18 / 68.82 |
| | SNN | 2 | W6A6/W8A8 | 69.06 / 69.93 | 74.23 / 76.45 | 44.88 / 45.90 | 72.98 / 72.90 | 75.68 / 78.45 | 67.37 / 68.73 |
| | **ES-TIF** | 2 | W6A6/W8A8 | 69.53 / 69.61 | 74.17 / 76.46 | 44.45 / 46.33 | 73.06 / 73.11 | 76.77 / 78.35 | 67.60 / 68.77 |
| | **ES-TDF** | 2 | W6A6/W8A8 | 68.35 / 68.75 | 73.33 / 76.33 | 45.14 / 45.65 | 73.40 / 73.91 | 76.22 / 78.29 | 67.29 / 68.58 |
| | SNN | 4 | W6A6/W8A8 | 69.46 / 69.85 | 74.11 / 76.59 | 45.14 / 46.16 | 72.98 / 73.40 | 76.82 / 78.24 | 67.70 / 68.85 |
| | **ES-TIF** | 4 | W6A6/W8A8 | 68.27 / 70.01 | 73.39 / 76.34 | 43.34 / 46.33 | 72.77 / 73.95 | 76.66 / 78.51 | 66.89 / 69.03 |
| **Llama 3 8B** | PrefixQ | - | W6A6/W8A8 | 71.82 / 73.09 | 77.61 / 78.96 | 50.94 / 53.75 | 75.59 / 77.99 | 77.69 / 80.47 | 70.73 / 72.85 |
| | DuQ | - | W6A6/W8A8 | 67.88 / 73.56 | 72.64 / 79.07 | 40.53 / 53.24 | 53.07 / 77.95 | 77.15 / 80.25 | 62.25 / 72.81 |
| | SNN | 1 | W6A6/W8A8 | 71.82 / 73.09 | 77.61 / 78.96 | 50.94 / 53.75 | 75.59 / 77.99 | 77.69 / 80.47 | 70.73 / 72.85 |
| | **ES-TIF** | 1 | W6A6/W8A8 | 74.11 / 73.72 | 77.40 / 79.04 | 49.40 / 53.41 | 75.59 / 77.65 | 77.75 / 80.36 | 70.85 / 72.84 |
| | **ES-TDF** | 1 | W6A6/W8A8 | 71.11 / 73.24 | 77.48 / 78.97 | 49.83 / 54.35 | 75.63 / 78.11 | 77.31 / 80.41 | 70.27 / 73.02 |
| | SNN | 2 | W6A6/W8A8 | 71.82 / 73.16 | 77.75 / 79.01 | 47.27 / 53.75 | 75.21 / 77.86 | 75.84 / 79.98 | 69.58 / 72.75 |
| | **ES-TIF** | 2 | W6A6/W8A8 | 72.22 / 73.24 | 77.58 / 78.83 | 48.89 / 53.50 | 75.63 / 77.57 | 77.86 / 80.03 | 70.44 / 72.63 |
| | **ES-TDF** | 2 | W6A6/W8A8 | 70.24 / 73.01 | 77.31 / 79.03 | 48.55 / 52.56 | 75.34 / 78.20 | 76.01 / 80.20 | 69.49 / 72.60 |
| | SNN | 4 | W6A6/W8A8 | 70.40 / 73.32 | 77.65 / 78.91 | 48.98 / 53.58 | 74.33 / 80.43 | 75.90 / 79.98 | 69.45 / 73.24 |
| | **ES-TIF** | 4 | W6A6/W8A8 | 71.19 / 73.09 | 77.34 / 78.81 | 48.81 / 53.75 | 74.37 / 77.90 | 76.77 / 80.30 | 69.69 / 72.77 |
| **Llama 3 70B** | PrefixQ | - | W8A8 | 79.32 | 85.65 | 62.37 | 82.79 | 84.11 | 78.85 |
| | DuQ | - | W8A8 | 80.82 | 84.83 | 63.48 | 85.73 | 84.39 | 79.85 |
| | SNN | 1 | W8A8 | 79.32 | 85.65 | 62.37 | 82.79 | 84.11 | 78.85 |
| | **ES-TIF** | 1 | W8A8 | 78.85 | 85.71 | 62.54 | 82.20 | 83.90 | 78.64 |
| | SNN | 2 | W8A8 | 79.48 | 85.70 | 62.88 | 82.87 | 83.90 | 78.97 |
| | **ES-TIF** | 2 | W8A8 | 79.08 | 85.60 | 62.88 | 82.62 | 83.90 | 78.82 |

### B.3 Additional Operation Counts Comparisons

To complement our analysis, we provide a breakdown of operation counts at the function level. In Figure 3, the results show that our ES-operators systematically replace multiply–accumulate (MAC) operations with accumulate (AC) and bit-shift operations. This change reflects a computation profile that is more naturally aligned with spiking accelerators, where ACs and shifts are much easier to support than dense MACs. Therefore, our method can be regarded as *spike-friendly* in terms of operator-level implementation.

Moreover, the scaling of SNN operations with time steps T, analyzed here with a fixed number of input tokens, reveals two distinct implementation strategies. As shown in Table 3, RMSNorm and Softmax perform their core, shift-based computation only once, resulting in a constant shift count regardless of T. Their ACs, however, grow linearly with T, reflecting the temporal cost of accumulation and differentiation. In contrast, SiLU executes its entire computation (both shifts and ACs) at every time step, causing its total operation count to scale linearly with T.

## C Ablation and Sensitivity Analysis

To further validate the robustness and efficiency of the proposed ES-operators, we conduct sensitivity studies at the operator level. In particular, we analyze the effect of the clipping interval length $2L$ and report how approximation errors vary with different ranges. Since our ES framework is a function-preserving approximation, replacing all nonlinear operators jointly is already a stricter test than replacing them individually. The fact that full-ES models retain accuracy therefore implies that each operator can be safely substituted without additional loss, and we omit redundant single-operator ablations.

Table 3: QNN vs. SNN Function-Level Operation Counts Comparison (G=$10^9$)

| Function | Operator | QNN Count (G) | SNN (T=1) Count (G) | SNN (T=2) Count (G) | SNN (T=4) Count (G) |
|---|---|---|---|---|---|
| RMSNorm | MACs | 0.1051 | 0.0000 | 0.0000 | 0.0000 |
| | ACs | 0.0524 | 0.0000 | 0.1049 | 0.3146 |
| | Shifts | 0.0000 | 0.3145 | 0.3145 | 0.3145 |
| SiLU | MACs | 2.3953 | 0.0000 | 0.0000 | 0.0000 |
| | ACs | 0.1409 | 0.1409 | 0.2818 | 0.5636 |
| | Shifts | 0.0000 | 1.4090 | 2.8180 | 5.6361 |
| Softmax | MACs | 0.6554 | 0.0000 | 0.0000 | 0.0000 |
| | ACs | 0.0406 | 0.4092 | 0.5321 | 0.7778 |
| | Shifts | 0.0000 | 0.4096 | 0.4096 | 0.4096 |

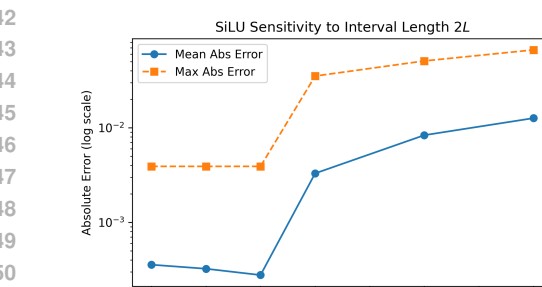

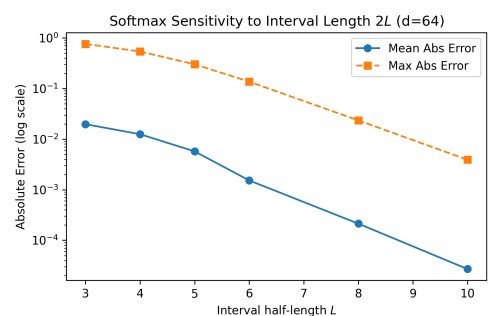

(a) ES-SiLU: sensitivity to interval length $2L$

(b) ES-Softmax ($d$=64): sensitivity to interval length $2L$

Figure 7: Operator-level sensitivity of ES-operators to the clipping interval length $2L$. (a) For SiLU, excessively large intervals cause the maximum error to increase rapidly. (b) For Softmax, larger intervals reduce errors, while small intervals ($L \leq 4$) lead to significant deviations. We therefore recommend $L$=5 as the default setting.

## C.1 FUNCTION SENSITIVITY ANALYSIS

**SiLU.** Figure 7(a) shows the sensitivity of ES-SiLU when $L$ is varied from 3 to 10. We observe that both mean and maximum approximation errors remain extremely small within moderate ranges (e.g., $L = 3, 4, 5$). However, when $L$ grows larger, the maximum error increases rapidly, reflecting the growing difficulty of approximating the extreme tails of the activation. This suggests that unnecessarily large intervals harm robustness without improving the error in the practically relevant region.

**Softmax.** In contrast, Figure 7(b) reports the sensitivity of ES-Softmax (fixed $d = 64$). Here, enlarging $L$ consistently reduces both mean and maximum errors, because clipping less aggressively preserves the exponential scaling inside the softmax. Nevertheless, small $L$ values ($\leq 4$) already induce non-negligible errors that may accumulate at the layer level.

**Recommended setting.** Taken together, these results justify our recommended default choice of $L = 5$ (i.e., interval $[-5, 5]$). This setting balances the two trends: it prevents extreme growth of SiLU errors while ensuring sufficiently accurate softmax evaluation. We therefore adopt $[-5, 5]$ as the standard interval for all experiments unless otherwise noted.

## LARGE LANGUAGE MODEL USAGE STATEMENT

In accordance with the ICLR 2026 Author Guidelines on the use of large language models, we acknowledge that large language models were used for phrasing refinement and grammar correction during manuscript preparation. All scientific ideas, algorithmic designs, and experimental results are solely the work of the authors.

