# OpenReview forum: "EasySpiking: Spike-Friendly Function Approximations for Spiking LLMs  Without Fine-Tuning"
_ICLR.cc/2026/Conference — Submitted to ICLR 2026_

### Official Review · Reviewer_opRL · 2025-10-27

**Soundness:** 2
**Presentation:** 2
**Contribution:** 2
**Rating:** 2
**Confidence:** 4

**Summary:**

This paper introduces EasySpiking, an ANN-to-SNN conversion framework that utilizes only simple arithmetic operations and requires no additional fine-tuning or retraining. It proposes the division neuron, the PWL-Exp unit, and the PolarNorm unit to replace floating-point division, exponential, and square root operations in ANN transformers.

**Strengths:**

1. The proposed EasySpiking framework relies solely on simple arithmetic operations, making it compatible with the spiking nature of SNNs.
2. This paper provides detailed theoretical analyses of the error bounds for the proposed EasySpiking submodules.

**Weaknesses:**

1. The main text lacks an analysis of computational efficiency. The QNN vs. SNN function-level operation counts comparison is placed in the appendix. I recommend including the efficiency analysis in the main text. Moreover, the proposed EasySpiking employs multi-bit spikes, which introduce additional computational overhead in spiking linear transformations. The comparison in Table 3 in the appendix overlooks this overhead. Additionally, Table 3 omits the overhead incurred by PolarNorm when accessing the lookup table.
2. Lack of comparison with state-of-the-art methods. This paper cites the state-of-the-art ANN-to-SNN language model conversion methods, such as SpikeZIP and Sorbet, but does not compare its performance against these approaches.
3. The comparison in Table 1 does not clearly highlight the advantages of the proposed method. I recommend adding comparisons of the number of operations or energy consumption.

Minors:

1. Figures 1 and 3 are too small and not vector graphics, making them difficult to read.
2. References are formatted inconsistently. For example, there are extra equal signs in the author names on line 587, and a missing conference name on line 593.

**Questions:**

I recommend conducting a comprehensive comparison with QNNs, direct SNN conversions, and state-of-the-art SNN language model conversion methods, while fully accounting for the additional overhead of multi-bit spikes, to better highlight the advantages of the proposed EasySpiking.

---

### Official Review · Reviewer_hf3D · 2025-10-29

**Soundness:** 2
**Presentation:** 1
**Contribution:** 2
**Rating:** 2
**Confidence:** 5

**Summary:**

The experiment in this article is not complete enough, and the expression is not clear enough.

**Strengths:**

This work introduces a family of spike-friendly approximations that collectively replace softmax, RMSNorm, and SwiGLU/SiLU.

**Weaknesses:**

The experiments in the paper are seriously insufficient:

1、The main text lacks comparisons with ANNs (LLaMA).

2、The main text lacks comparisons with QNNs (Quantized LLaMA).

3、Lack of power consumption and inference speed estimates.

4、The task is singular, and ANN2SNN lacks multi-task evaluation, such as mathematical reasoning(MATH, GSM8k), code generation(HumanEval, MBPP), Massive Multitask Language Understanding(MMLU), etc.

5、The expression of this paper is unclear. For example, the footnotes in the only table in the main text are unclear. What does W6A6 mean, and what is its relationship with SNN? Why is it an SNN?

6、Rope lacks discussion; rope is also a nonlinear operation.

7、Lack of a neuromorphic chip verification discussion, such as Intel Loihi, IBM TrueNorth, etc.

**Questions:**

see above

---

### Official Review · Reviewer_oq8u · 2025-10-30

**Soundness:** 2
**Presentation:** 1
**Contribution:** 2
**Rating:** 2
**Confidence:** 4

**Summary:**

This paper proposes EasySpiking, a modular framework for converting ANN-based LLMs into spike-friendly architectures without fine-tuning.
The authors introduce three modules, a Division Neuron, a Piecewise Linear Exponential (PWL-Exp), and a PolarNorm (CORDIC-based) approximation to replace non-spike-compatible operators such as SiLU, Softmax, and RMSNorm.
They provide function-level, module-level, and model-level evaluations.

**Strengths:**

1. The use of CORDIC for approximating RMSNorm denominators is practically relevant for neuromorphic hardware.
2. The paper systematically evaluates its approximations on function, module, and model levels.
3. The author theoretically proved their operations has bounded approximation error.

**Weaknesses:**

1. The claim that integer-valued spikes remain “spike-like” is unsubstantiated. SpikeZIP-TF uses spikes in {+1, 0, –1}, which preserves binary emission semantics. In contrast, EasySpiking keeps arbitrary integers without proving they are bounded or sparse enough to be energy-efficient. This design makes the resulting network arguably non-SNN in spirit.
2. The authors claim “no lookup table” in abstract, yet lookup table are used in the method.
3. SorBET is a BERT-like model that uses GELU and LayerNorm, not SiLU or RMSNorm. Therefore, comparing the function-level error of ES-SiLU and ES-RMSNorm to SorBET’s approximations is conceptually meaningless.
4. The paper lacks a proper Conclusion section, and there is no energy or latency analysis of the full model to demonstrate whether EasySpiking actually improves efficiency which is the core motivation of SNN design.
5. Several paragraphs are confusing or repetitive, see “Questions”.

**Questions:**

1. If spikes are integer-valued rather than binary, can they still achieve energy savings compared to quantized ANNs? How are these integers represented in neuromorphic hardware? Remark 1 cites SpikeZIP and SorBET as precedent for integer spikes, but neither uses multi-bit integer spikes. This appears factually incorrect.
2. In this paper, each operator involves a Division Neuron. Is such discrete division truly neuromorphic-friendly, given SNNs are event-driven and additive by nature?
3. In Remark 2, where does the threshold ε_exp < 10⁻² come from? Even if the bound is chosen, it only defines a relation between L and K, not specific values like L = 5, K = 64. It seems this condition merely rationalizes the chosen example rather than deriving it.
4. The authors claim Sorbet’s approximation error grows with dimensionality, yet in Figure 3(b) it does not visibly increase. Please clarify.

---

### Official Review · Reviewer_3KJR · 2025-10-30

**Soundness:** 2
**Presentation:** 2
**Contribution:** 2
**Rating:** 2
**Confidence:** 3

**Summary:**

The paper introduces spike friendly approximations for softmax, SiLU and RMSNorm.

**Strengths:**

There is indeed a need to have SNN-friendly approximations for these functions in order that SNNs can be used on LLMs.

**Weaknesses:**

There is a major issue in the evaluation: Sorbet is a BERT-like model that uses GELU and LayerNorm, not SiLU or RMSNorm. Therefore, comparing the function-level error of ES-SiLU and ES-RMSNorm to Sorbet’s approximations is conceptually meaningless.

I also took a look at the code and as far as I can make out, it just replaces some of the functions in the Llama models with what the authors'. But it is unclear how the entire model can be put together. For example, when computing self-attention, is QK computed as spikes before softmax and then somehow (and this is not clear to me) multiplied with V? I cannot make out from the description how an entire self-attention block can be composed using their function.

Also, the experiments seem to say that T=1. If so, I am not sure how the divide neuron or the piece-wise function approximation can be done. How long will a single T be?

The paper has left me with too many unanswered questions.

**Questions:**

1. How were your SiLU and RMSnorm for Sorbet obtained?

2. Are the experiments performed on entirely SNN-ized equivalent of Llama?

3. How does the divide neuron work when T = 1?

**Details Of Ethics Concerns:**

None.

---

### Meta-Review · Area_Chair_12Xr · 2026-01-06

**Summary:**

This paper presents a training-free ANN-to-SNN conversion method for LLMs. To this end, the paper introduces a family of spike-friendly approximations of transformer nonlinearities, such as softmax, RMSNorm, and SiLU. The paper integrates the proposed approximation method on LLaMA models.

Reviewers raised major concerns about the proposed method and evaluation (e.g., mismatch of the claimed and actual nonlinearities, usage of lookup tables, insufficient evaluation, etc.). The authors did not submit a rebuttal. Overall, I agree with the reviewers and think the paper needs major revisions to meet the bar of ICLR.

**Reviewer Concerns:**

Reviewers raised major concerns about the proposed method and evaluation (e.g., mismatch of the claimed and actual nonlinearities, usage of lookup tables, insufficient evaluation, etc.). The authors did not submit a rebuttal.

**Reviewer Scores:**

Since the authors did not submit a rebuttal, no discussion took place and all reviewer scores remained the same.

---

### Decision · Program_Chairs · 2026-01-26

Reject